# Sociodemographic characteristics and risk factors related to SARS-CoV-2 infection in Luanda, Angola

**Cruz S. Sebastião**[1,2,3], **Zoraima Neto**[2]*, **Pedro Martinez**[2], **Domingos Jandondo**[2], **Janete Antonio**[2], **Manuela Galangue**[2], **Marcia de Carvalho**[2], **Kumbelembe David**[2], **Julio Miranda**[2], **Pedro Afonso**[2], **Luzia Inglês**[2], **Raisa Rivas Carrelero**[2], **Jocelyne Neto de Vasconcelos**[1,2]*, **Joana Morais**[2,4]*

1 Centro de Investigação em Saúde de Angola, Caxito, Angola, 2 Instituto Nacional de Investigação em Saúde, Luanda, Angola, 3 Instituto Superior de Ciências da Saúde, Universidade Agostinho Neto, Luanda, Angola, 4 Faculdade de Medicina, Universidade Agostinho Neto, Luanda, Angola

* jfm.morais9@gmail.com (JM); zoraima.neto@gmail.com (ZN); jocelyne.vasconcelos@cisacaxito.org (JNV)

## Abstract

This study aimed to investigate the characteristics related to SARS-CoV-2 in Luanda, Angola. A total of 622 individuals were screened for SARS-CoV-2 from January to September 2020. Chi-square and logistic regression were used to identify the relationship between sociodemographic characteristics and SARS-CoV-2. Of the 622 tested, 14.3% tested positive. The infection rate was the same for both genders (14.3%). Individuals ≥40 years old, from non-urbanized areas, and healthcare professionals had a higher frequency of infection. The risk of infection was very high in individuals ≥60 years old (AOR: 23.3, 95% CI: 4.83–112), in women (AOR: 1.24, 95% CI: 0.76–2.04), in Luanda (AOR: 7.40, 95% CI: 1.64–33.4), and healthcare professionals (AOR: 1.27, 95% CI: 0.60–2.71), whereas a low risk was observed in individuals from urbanized areas (AOR: 0.44, 95% CI: 0.26–0.75). Our results suggest that Angolan authorities should implement a greater effort in non-urbanized areas and among healthcare professionals since when these individuals presented any indication for a COVID-19 test, such as fever/cough/myalgia, they were more likely to test positive for SARS-CoV-2 than having some other cause for symptoms.

## Introduction

The severe acute respiratory syndrome of coronavirus 2 (SARS-CoV-2) is a new member of the family *Coronaviridae* [1, 2]. This new coronavirus was initially identified in Wuhan, China, in December 2019. The SARS-CoV-2 infection causes coronavirus disease-2019 (COVID-19) [3]. Infected individuals may experience fever, dry cough, fatigue, shortness of breath, myalgia, diarrhea, and in severe cases, the SARS-CoV-2 infection causes viral pneumonia and leads to death [4–6]. By the end of October 2020, over 40 million infections and over a million deaths related to SARS-CoV-2 infection have been reported globally [7]. In Africa, the pandemic is increasing in the number of infections and deaths [7, 8].

**Data Availability Statement:** All relevant data are within the paper.

**Funding:** The authors received no specific funding for this work.

**Competing interests:** The authors have declared that no competing interests exist.

The first cases of SARS-CoV-2 infection in Angola were detected in March 2020 at the Instituto Nacional de Investigação em Saúde (INIS), the national reference biomedical research institute located in Luanda, the capital city of Angola. By the end of October 2020, about 10 000 infections and 270 deaths related to SARS-CoV-2 infection had been reported [7]. At the beginning of the COVID-19 pandemic in Angola, the INIS was the only institution responsible for laboratory testing and surveillance of SARS-CoV-2 cases in the country.

Angola is a country in sub-Saharan Africa with more than 25 million inhabitants living in 18 provinces, with about 48% male inhabitants and 52% female, with an average age of 20.6 years. Luanda province is the most inhabited with about 7 million inhabitants [9]. A large proportion of the population from Luanda province lives in slums with poor basic sanitation and limited access to health care [10]. In addition, because of the oil trade, international business travelers have intensely visited the Luanda province, which could increase the chance of importing SARS-CoV-2 to the country's capital and easily spread to other regions. At present, there is little information published on the sociodemographic characteristics, as well as, the risk factors associated with the emergence and spread of the SARS-CoV-2, especially in low- and middle-income countries (LMICs), such as Angola [11]. However, in this study, we investigate in detail the sociodemographic characteristics including risk factors related to the emergence and spread of the SARS-CoV-2 infection in Luanda, the capital city of Angola.

## Materials and methods

### Study design and participants

This is a cross-sectional study performed with 622 individuals out of 16 028 individuals tested for the SARS-CoV-2 infection between January to September 2020 at Instituto Nacional de Investigação em Saúde (INIS), located in Luanda, the capital city of Angola. The INIS is an Angolan institute of scientific research directly subordinated and supported by the Angolan Ministry of Health, whose main objective is to generate, develop and disseminate scientific, technological, and strategic knowledge about health and its determinants, for strengthening public health policies and improving the national health system in Angola (http://www.inis. ao/index.php/institucional/o-instituto). The sociodemographic (age, gender, province, residence area, and occupation) and clinical characteristics (signs/symptoms and comorbidities at admission) of participants were collected by the research team using a structured questionnaire for the surveillance and investigation of SARS-CoV-2 cases. We did not develop a questionnaire as part of this study. The questionary used in the study was prepared and made available by the national public health directorate of Angola. However, the questionnaire was not validated prior to testing on study participants, since the questionnaire is validated by the national public health directorate of Angola for the surveillance and investigation of SARS-- CoV-2 cases in Angola. The study protocol was reviewed by the scientific coordination of INIS and ethical approval was obtained from the national ethics committee of the Ministry of Health of Angola (nr.25/2020). The participants or legal guardians of minors were informed about the study and verbal consent was secured before being enrolled in the study. The information from the studied population was fully anonymized, used for the stated objectives, and kept confidential in the INIS.

### Sample collection and laboratory testing

Smears or swabs from the upper respiratory tract were obtained in all participants during admission using the Viral Transport Media (VTM) 3 mL with a minitip flocked swab from the KaiBili (Hangzhou Genesis, China). People were tested for the following reasons as follows: if they had presumably COVID-19 suspicious syndrome; if they have been in contact with

someone infected; or if they traveled to any country or region with active transmission of SARS-CoV-2. The smears or swabs were stirred in the transport solution to elute the cells that adhere to the swabs. Then, the swabs were kept in a sterile viral transport medium and transported immediately to the molecular biology laboratory of INIS. The viral ribonucleic acid (RNA) was manually extracted from 200μL of the liquid specimen using the Nucleic Acid Isolation or Purification Reagent produced by Da An Gene Co., Ltd. of Sun Yat-sen University (Da An Gene, China). The SARS-CoV-2 infection was screened and confirmed with real-time reverse transcriptase-polymerase chain reaction (RT-PCR) assay using the Applied Biosystems 7500 Fast RT-PCR System (Thermo Fisher Scientific), in the molecular biology laboratory of INIS, using a protocol previously described for the detection of 2019 novel Coronavirus (2019-nCoV) RNA (PCR-Fluorescence Probing) (Da an Gene, China) [12]. Briefly, the RT-PCR was carried out using 5μL of the extracted RNA from liquid specimen to each reaction tube in a final reaction volume of 25μL that contain specific primers and fluorescent probes targeting *in vitro* qualitative detection of SARS-CoV-2 ORF1ab and N genes. Cycling conditions consisted of 15 minutes at 50˚C for reverse transcription and 15 minutes at 95˚C for pre-denaturation, followed by 45 cycles of 15 seconds at 94˚C and 45 seconds at 55˚C, for nucleic acid amplification and fluorescence detection. The fluorescent dyes FAM, VIC, and Cy5 were used to detect light emissions. Positive and negative control samples were included for each RT-PCR assay. Specimens with cycle threshold (Ct) up to 40 were considered positive for SARS-CoV-2 infection, whereas specimens with Ct value above 40 or without Ct value for the FAM and VIC dyes were considered negative for SARS-CoV-2 infection.

### Statistical analysis

The data were coded and analyzed using SPSS version 25 (IBM SPSS Statistics, USA). The normality of data distribution was checked using the values of skewness and kurtosis. Categorical variables were presented as frequencies and percentages, while continuous variables with the data normally distributed were presented as mean and standard deviation (SD). Chi-square ($X^2$) test was used to compare frequencies and identify the relationship between categorical variables. Besides that, logistic regression analysis and odds ratio (OR) with their 95% confidence intervals (CIs) were calculated to determine the strength and direction of the interaction between variables. The Hosmer-Lemeshow test was used to check the quality of the model fit. All values shown are two-tailed and were considered statistically significant when p<0.05.

### Results

Table 1 summarizes the sociodemographic characteristics and risk factors related to SARS--CoV-2 infection in Luanda, Angola. A total of 622 individuals out of 16 028 individuals tested for SARS-CoV-2 from January to September 2020 completed all sociodemographic and clinical data and were included in the analysis. Angola has a weak surveillance system in which lack of epidemiological data is a major problem that being the reason why only 622 individuals with complete epidemiological data were included in this paper. Of these, 244/622 (39.2%) were female and 378/622 (60.8%) were male. The age range varied between 1–92 years old, with an average of 32.3 ± 18.7. The study was predominated by adults aged 30–39 years (23.0%, 143/622), followed by individuals aged 40–49 years (16.2%, 101/622). Moreover, individuals from Luanda province (93.9%, 584/622), living in an urbanized area (75.1%, 467/622), and unemployed (73.0%, 454/622), were also predominant. A total of 89/622 (14.3%) of the studied population tested positive for RT-PCR against SARS-CoV-2. All individuals who tested positive by RT-PCR were placed in quarantine centers established by the Angolan Ministry of Health, for clinical follow-up and isolation. Upon entering quarantine in most cases we

**Table 1. Sociodemographic characterization and risk factors of SARS-CoV-2 infection in Luanda, Angola.**

| Characteristics | n (%) | Test positivity to SARS-CoV-2 | | | Univariate analysis | | Multivariate analysis | |
|---|---|---|---|---|---|---|---|---|
| | | No (%) | Yes (%) | p-value | OR (95% CI) | p-value | AOR (95% CI) | p-value |
| Overall | 622 (100) | 533 (85.7) | 89 (14.3) | | | | | |
| Age groups | | | | | | | | |
| <10y | 97 (15.6) | 95 (97.9) | 2 (2.1) | **<0.001** | 1 | - | 1 | - |
| 10-19y | 67 (10.8) | 65 (97.0) | 2 (3.0) | | 1.46 (0.20–10.6) | 0.708 | 1.65 (0.23–12.1) | 0.624 |
| 20-29y | 98 (15.8) | 83 (84.7) | 15 (15.3) | | 8.58 (1.90–38.7) | **0.005** | 9.78 (2.13–44.9) | **0.003** |
| 30-39y | 143 (23.0) | 117 (81.8) | 26 (18.2) | | 10.6 (2.44–45.6) | **0.002** | 11.9 (2.30–52.2) | **0.001** |
| 40-49y | 101 (16.2) | 86 (85.1) | 15 (14.9) | | 8.29 (1.84–37.3) | **0.006** | 9.23 (2.01–42.3) | **0.004** |
| 50-59y | 73 (11.7) | 56 (76.7) | 17 (23.3) | | 14.4 (3.21–64.8) | **<0.001** | 14.7 (3.20–67.2) | **0.001** |
| ≥60y | 43 (6.9) | 31 (72.1) | 12 (27.9) | | 18.4 (3.90–86.7) | **<0.001** | 23.3 (4.83–112) | **<0.001** |
| Gender | | | | | | | | |
| Female | 244 (39.2) | 209 (85.7) | 35 (14.3) | 0.984 | 1.01 (0.64–1.59) | 0.984 | 1.24 (0.76–2.04) | 0.390 |
| Male | 378 (60.8) | 324 (85.7) | 54 (14.3) | | 1 | - | 1 | - |
| Province | | | | | | | | |
| Outside Luanda | 38 (6.1) | 36 (94.7) | 2 (5.3) | 0.100 | 1 | - | 1 | - |
| Luanda | 584 (93.9) | 497 (85.1) | 87 (14.9) | | 3.15 (0.75–13.3) | 0.119 | 7.40 (1.64–33.4) | **0.009** |
| Place of residence | | | | | | | | |
| Rural area | 155 (24.9) | 125 (80.6) | 30 (19.4) | **0.038** | 1 | - | 1 | - |
| Urban area | 467 (75.1) | 408 (87.4) | 59 (12.6) | | 0.60 (0.37–0.98) | **0.040** | 0.44 (0.26–0.75) | **0.002** |
| Occupation | | | | | | | | |
| Unemployed | 454 (73.0) | 396 (87.2) | 58 (12.8) | 0.149 | 1 | - | 1 | - |
| Healthcare professionals | 51 (8.2) | 40 (78.4) | 11 (21.6) | | 1.88 (0.91–3.87) | 0.087 | 1.27 (0.60–2.71) | 0.529 |
| Others | 117 (18.8) | 97 (82.9) | 20 (17.1) | | 1.41 (0.81–2.45) | 0.227 | 1.14 (0.63–2.05) | 0.673 |

[a]Adjusted for all the explanatory variables listed.

Bold results mean they were significant in the chi-square or logistic regression (p<0.05).

lost track of patients and it was not possible to do a follow-up of patients, to obtain the result of the disease severity, and the clinical outcome. The positivity rate by SARS-CoV-2 was the same for both genders (14.3%), while individuals over 40 years old, from rural areas in Luanda, and individuals involved in health care had a higher frequency of infection by SARS-CoV-2. Age and place of residence were statistically related to SARS-CoV-2 infection (p<0.05), while, gender, province, and occupation showed no relationship (p>0.05) with the SARS-CoV-2 infection. Besides that, our results showed that the risk for SARS-CoV-2 infection was higher in individuals with age equal or over 60 years [AOR: 23.3 (95% CI: 4.83–112), p<0.001], in women [AOR: 1.24 (95% CI: 0.76–2.04), p = 0.390], in individuals from Luanda province [AOR: 7.40 (95% CI: 1.64–33.4), p = 0.009], and in healthcare professionals [AOR: 1.27 (95% CI: 0.60–2.71), p = 0.529]. On the other hand, the risk of infection was lower in individuals from urbanized areas of Luanda [AOR: 0.44 (95% CI: 0.26–0.75), p = 0.002]. Moreover, our results also revealed that almost all (98.9%, 88/89) individuals tested for SARS-CoV-2 infection at INIS, had symptoms related to COVID-19. There was only one individual who tested positive for SARS-CoV-2 infection who was listed as asymptomatic. Even so, no significant relationship was observed (p>0.05) between sociodemographic characteristics and the worsening of the infection (Table 2). Cough (65.9%, 58/88), fever (43.2%, 38/88), headache (26.1%, 23/88), shortness of breath (18.2%, 16/88), malaise (17.0%, 15/88), and sore throat (12.5%, 11/88), were the most frequent signs and symptoms presented by the COVID-19 patients in Luanda

**Table 2. Sociodemographic characteristics related to disease severity among patients infected with SARS-CoV-2 in Luanda, Angola.**

| Characteristics | n (%) | Symptomatic disease | | |
|---|---|---|---|---|
| | | No (%) | Yes (%) | p-value |
| Overall | 89 (100) | 1 (1.1) | 88 (98.9) | |
| Age groups | | | | |
| <10y | 2 (2.2) | 0 (0.0) | 2 (100) | 0.545 |
| 10-19y | 2 (2.2) | 0 (0.0) | 2 (100) | |
| 20-29y | 15 (16.9) | 1 (6.7) | 14 (93.3) | |
| 30-39y | 26 (29.9) | 0 (0.0) | 26 (100) | |
| 40-49y | 15 (16.9) | 0 (0.0) | 15 (100) | |
| 50-59y | 17 (19.1) | 0 (0.0) | 17 (100) | |
| ≥60y | 12 (13.5) | 0 (0.0) | 12 (100) | |
| Gender | | | | |
| Female | 35 (39.3) | 1 (2.9) | 34 (97.1) | 0.212 |
| Male | 54 (60.7) | 0 (0.0) | 54 (100) | |
| Province | | | | |
| Outside Luanda | 2 (2.2) | 0 (0.0) | 2 (100) | 0.879 |
| Luanda | 87 (97.8) | 1 (1.1) | 86 (98.9) | |
| Place of residence | | | | |
| Rural area | 30 (33.7) | 0 (0.0) | 30 (100) | 0.473 |
| Urban area | 59 (66.3) | 1 (1.7) | 58 (98.3) | |
| Occupation | | | | |
| Unemployed | 58 (65.2) | 1 (1.7) | 57 (98.3) | 0.763 |
| Healthcare professionals | 11 (12.4) | 0 (0.0) | 11 (100) | |
| Others | 20 (22.5) | 0 (0.0) | 20 (100) | |

(Table 3). Interestingly, we observed a relationship between Ct values and myalgia or arthralgia (p = 0.041) (Table 3).

## Discussion

The results on COVID-19 obtained in this study seem to be valuable data that should be shared, especially since they are from under-resourced areas and are important for the present COVID-19 situation. To the best of our knowledge, this is the first study that details the socio-demographic aspects of the SARS-CoV-2 infection in Luanda, the epicenter of the COVID-19 pandemic in Angola. In this study, the positive rate of SARS-CoV-2 infection was 14.3%. The positive rate was the same for both men and women and no relationship was observed between gender and vulnerability to SARS-CoV-2 infection (p>0.05). Previous studies in Wuhan, China, found a higher positive rate of SARS-CoV-2 infection in men compared to women [13]. Besides that, previous severe acute respiratory syndrome (SARS) and middle east respiratory syndrome (MERS) pandemics were also mostly observed in men compared to women [14–16]. Some studies reported that women have reduced susceptibility to contracting SARS-CoV-2 due to the protection provided by the activity of X-linked genes where several genes that encode molecules of the innate immune system are located and sex-specific steroids which modulate the innate and adaptive immune response against viral infections [16, 17]. However, our results even though no statistical significance indicated that the likelihood to test positive to SARS-CoV-2 infection could be higher in women (AOR: 1.24, p = 0.390) compared to men (Table 1). These results could indicate that there may be sex-dependent

**Table 3. Clinical characteristics at admission and their relationship with Ct value among COVID-19 patients in Luanda, Angola.**

| Signs, symptoms, and comorbidities | All patients (n = 88) | Ct value (SARS-CoV-2 N gene) | | |
| --- | --- | --- | --- | --- |
| | | Ct ≤ 30 (%) | Ct > 30 (%) | p-value |
| Cough | 58 (65.9%) | 19 (32.8) | 39 (67.2) | 0.358 |
| Fever (temperature ≥37.3˚C) | 38 (43.2%) | 14 (36.8) | 24 (63.2) | 0.191 |
| Headache | 23 (26.1%) | 8 (34.8) | 15 (65.2) | 0.522 |
| Shortness of breath | 16 (18.2%) | 6 (37.5) | 10 (62.5) | 0.441 |
| Malaise | 15 (17.0) | 7 (46.7) | 8 (53.3) | 0.111 |
| Sore throat | 11 (12.5%) | 5 (45.5) | 6 (54.5) | 0.216 |
| Runny nose | 10 (11.4%) | 3 (30.0) | 7 (70.0) | 0.973 |
| Diarrhea | 6 (6.8%) | 3 (50.0) | 3 (50.0) | 0.255 |
| Comorbidities | 5 (5.7%) | 3 (60.0) | 2 (40.0) | 0.124 |
| Diabetes | 1 (1.1%) | 0 (0.0) | 1 (100) | 0.515 |
| Tuberculosis | 1 (1.1%) | 1 (100) | 0 (0.0) | 0.120 |
| Cardiac disease | 1 (1.1%) | 1 (100) | 0 (0.0) | 0.120 |
| Myalgia or arthralgia | 4 (4.5%) | 3 (75.0) | 1 (25.0) | **0.041** |
| Nausea or vomiting | 4 (4.5%) | 1 (25.0) | 3 (75.0) | 0.838 |
| Joint pain | 3 (3.4%) | 0 (0.0) | 3 (100) | 0.254 |
| Chest pain | 3 (3.4%) | 0 (0.0) | 3 (100) | 0.254 |
| Chills | 2 (2.3%) | 0 (0.0) | 2 (100) | 0.354 |
| Fatigue | 1 (1.1%) | 1 (100) | 0 (0.0) | 0.120 |

The bold result mean that was significant in the chi-square test (p<0.05).

differences in the outcomes of SARS-CoV-2 infection, which emphasizes that gender might be a biological and social variable that should be considered in controlling the emergence and spread of viral infectious diseases in Luanda, the capital city of Angola [18].

A strong relationship between age and susceptibility to test positive to SARS-CoV-2 infection was observed in our study (p<0.001) (Table 1). Although positive cases of SARS-CoV-2 infection have been reported in patients of all ages, a higher positive rate of SARS-CoV-2 infection was observed in adults (Table 1), which could show that in Luanda, there is a greater susceptibility for adult individuals to test positive for SARS-CoV-2 infection, compared to young individuals. On the other hand, it seems that younger individuals were more likely to be tested for symptoms that were not caused by COVID-19 than older individuals. We can also say that perhaps younger individuals had easier access to the tests, therefore, they had more mild and nonspecific symptoms. The positive rate of individuals under 20 years old was very low and varied between 2.1%–3.0%, while in individuals aged 20 to 30 years ranged from 15.3%–18.2%, individuals aged 40 to 50 years ranged from 14.9%–23.3%, and in adults age over 60 years and over was 27.9%. The individuals aged 60 or over (AOR: 23.3, p<0.001) were more likely to test positive for SARS-CoV-2 infection compared to the group of young individuals under the age of 60. Our results agree with previous studies that found a high positive rate of SARS-CoV-2 infection in older patients from China, Italy, Spain, Germany, Netherlands, and Canada [19]. One of the reasons for adults or older adults to be the most vulnerable group to test positive for SARS-CoV-2 infection could be the fact that adults have weaker immune functions because immunosenescence increases with age. COVID-19 positivity rates may also be influenced by chronic diseases such as hypertension and diabetes, which are conditions that could worsen SARS-CoV-2 infection [20]. In fact, previous studies have shown a greater risk of viral infection and worse outcomes related to viral infection in older adults and adults who have certain comorbidities [21, 22].

Knowledge of sociodemographic characteristics that constitute the greatest risk of spread viral infectious disease in the community must be studied and presented to strengthen decision-making and ensure greater awareness about the COVID-19 pandemic in the community. Our study showed that Luanda province might be an independent predictor for SARS-CoV-2 infection (OR: 3.15, p = 0.119). Moreover, when adjusted with age, gender, place of residence, and occupation, the risk of SARS-CoV-2 infection in Luanda was 7.4 times (95% CI: 1.64–33.4, p = 0.009), compared to the infection rate outside Luanda. On the other hand, unlike other studies that did not show a relationship between place of residence and susceptibility to SARS-CoV-2 infection, our results showed that the place of residence (rural or urbanized area) may be an important factor in the spread of viral infectious disease or SARS-CoV-2 infection in the community. Individuals from urbanized areas seem to have less risk to test positive for SARS-CoV-2 infection (AOR: 0.44, p = 0.002), compared to individuals from rural areas (Table 1). These results were similar to those reported in Brazil, where a lower risk of infection was observed in the country's capital compared to other regions [23]. The risk of infection by SARS-CoV-2 amongst healthcare professionals could be high (AOR: 1.27, p = 0.529), compared to the unemployed and employees from other sectors (Table 1). At this time, healthcare professionals face considerable physical and mental stress, stigma, and pain for losing patients and healthcare colleagues who acquired SARS-CoV-2 infection or died of COVID-19 [8].

Currently, the world is facing a global pandemic of SARS-CoV-2, with many countries experiencing a second wave of infections [24]. In Angola, there is a continuous trend of an increase in the number of infections and deaths related to COVID-19 [7, 25]. Community transmission and increased screening capacity for SARS-CoV-2 can help explain the growing number of infected and deaths related to COVID-19 in Angola. The same pattern has been noticed in the African region, mainly in South Africa, Ethiopia, Kenya, and Botswana which also have community transmission and notable increases in cases of infection and deaths related to COVID-19 [7]. According to the COVID-19 daily update bulletin from the Ministry of Health of Angola, the COVID-19 pandemic in Luanda was asymptomatic in the first three months with positive patients reporting no symptoms [25], however, as the pandemic progressed more cases were linked to symptoms such as cough (65.9%), fever (43.2%), headache (26.1%), shortness of breath (18.2%), malaise (17.0%), and sore throat (12.5%) (Table 3). Our results are similar to previous studies who also reported a higher frequency of fever and cough in individuals infected with SARS-CoV-2 [4–6]. On the other hand, previous studies have shown high viremia after the virus enters the human body and that the main clinical manifestations resulting from high viral load are fever, sore throat, shortness of breath, fatigue, and diarrhea [26]. Besides that, previous studies also observed a higher frequency of clinical manifestations such as fever, cough, myalgia, fatigue, and diarrhea, in patients who died due to COVID-19 [4, 5]. Almost all individuals infected with SARS-CoV-2 in our study had high Ct values for the SARS-CoV-2 N gene, which could indicate low viremia and mild or moderate disease. On the other hand, most patients with diarrhea, comorbidities, myalgia, and fatigue, had low Ct values, which could indicate high viremia and severe disease. Interestingly, low Ct values were significantly associated with myalgia or arthralgia which may be associated with high viremia or severe disease (p<0.05) (Table 3). Therefore, our results showed that the presence of myalgia or arthralgia might be an indication of high viremia or severe disease related to SARS-CoV-2 infection. However, the relationship between clinical manifestations related to SARS-CoV-2 infection and the Ct values or viral load should be further studied to help healthcare professionals intervene immediately and to avoid unfavorable clinical outcomes. The high number of individuals detected with symptomatic infection in Luanda may be an indication that the population of Luanda is mostly looking for healthcare when the disease is at a severe stage. However, these data may not reflect the current state of the COVID-19 pandemic in

Luanda, or in other provinces of Angola, since individuals tested in other health units responding to the COVID-19 pandemic in Angola were not included. Moreover, there has been a big gap in the country's pandemic response capacity, mainly due to the lack of human resources and limited access to protective equipment for healthcare professionals, COVID-19 patients, and the general population. Containment measures such as the regular use of masks, the provision of soap and water for handwashing, and physical distance, these are low-cost interventions, but they represent major challenges in urbanized and non-urbanized regions of Luanda, the capital city of Angola. Therefore, failure to comply with the containment measures could lead to increased spread and deaths related to SARS-CoV-2 infection in Angola. On the other hand, the high exposure of healthcare professionals and the lack of protective equipment can be a crucial factor for increased SARS-CoV-2 infection and mortality amongst healthcare professionals and their families [8]. However, we strongly suggest that the Ministry of Health of Angola should pay special attention to the safety of healthcare professionals, as well as their mental wellbeing. Moreover, increasing the number of professionals in health units in Luanda, as well as increasing access and availability of personal protective equipment, especially for healthcare professionals, must be guaranteed for their protection and to increment the response capacity of the COVID-19 pandemic in Angola. Furthermore, studies on the mental health of healthcare professionals during the COVID-19 pandemic should be carried out in Angola, to help define strategies to improve the response capacity.

The current study has potential limitations. Firstly, the study included only individuals tested for SARS-CoV-2 at INIS. Secondly, the data of the individuals included in this study may not represent the whole population of Luanda or other provinces of Angola. Thirdly, with the lack of data on disease severity and the limited number of participants infected with SARS-CoV-2, it is difficult to assess disease severity and mortality. Finally, more detailed patient information, specifically SARS-CoV-2 viral load, length of stay in the hospital or quarantine centers, and clinical outcomes were not available at the time of analysis. Despite these limitations, this study allows for a primary assessment of the sociodemographic characteristics of the SARS-CoV-2 infection in Angola. However, future studies including more participants with more clinic and sociodemographic information, especially the viral load, length of stay in the hospital, and clinical outcome, should be carried out, to reinforce ongoing strategies to combat the COVID-19 pandemic in Angola. Besides that, retrospective studies capable of determining the exact period of introduction of SARS-CoV-2 in Angola must be carried out. On the other hand, genetic characterization of SARS-CoV-2 strains and the relationship between SARS-CoV-2 with vector-borne diseases should be explored, since the Luanda province is endemic in vector-borne diseases, especially dengue, zika, chikungunya, and malaria [27, 28].

## Conclusions

Our results showed that age and other sociodemographic characteristics are important factors for the emergence and spread of SARS-CoV-2 infection in Luanda, the capital city of Angola. A greater effort to control the COVID-19 pandemic should be implemented in people from non-urbanized areas and among healthcare professionals, since when these individuals presented with an indication for COVID-19 testing such as cough, fever, and myalgias, that they were more likely to test positive for SARS-CoV-2 rather than have some other cause for their symptoms. Further clinical and epidemiological studies are needed, for the deeper characterization of the groups most vulnerable to SARS-CoV-2 infection in Angola. In addition, these studies will reinforce the information utilized for decision-making about the ongoing strategies to control the COVID-19 pandemic in Angola.

## Acknowledgments

Thanks to the Ministry of Health of Angola and partners to logistic support. Moreover, thank all the individuals from Luanda who participated freely in this study. Thanks to the research team of INIS and CISA for the technical support and data collection, and thank Joana Sebastião for scientific and logistic support.

## Author Contributions

**Conceptualization:** Cruz S. Sebastião, Zoraima Neto, Jocelyne Neto de Vasconcelos, Joana Morais.

**Data curation:** Cruz S. Sebastião, Zoraima Neto, Jocelyne Neto de Vasconcelos, Joana Morais.

**Formal analysis:** Cruz S. Sebastião, Zoraima Neto, Jocelyne Neto de Vasconcelos, Joana Morais.

**Investigation:** Cruz S. Sebastião, Pedro Martinez, Domingos Jandondo, Janete Antonio, Manuela Galangue, Marcia de Carvalho, Kumbelembe David, Julio Miranda, Pedro Afonso, Luzia Inglês, Raisa Rivas Carrelero.

**Methodology:** Cruz S. Sebastião.

**Project administration:** Zoraima Neto, Jocelyne Neto de Vasconcelos, Joana Morais.

**Supervision:** Zoraima Neto, Pedro Martinez, Jocelyne Neto de Vasconcelos, Joana Morais.

**Validation:** Cruz S. Sebastião, Zoraima Neto, Jocelyne Neto de Vasconcelos, Joana Morais.

**Writing – original draft:** Cruz S. Sebastião.

**Writing – review & editing:** Cruz S. Sebastião, Zoraima Neto, Jocelyne Neto de Vasconcelos, Joana Morais.

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
