## [Decision Letter · Decision Letter 0]

4 Feb 2021

PONE-D-20-36563

Sociodemographic characteristics and risk factors related to SARS-CoV-2 infection in Luanda, Angola

PLOS ONE

Dear Dr. dos Santos Sebastiao,

Thank you for submitting your manuscript to PLOS ONE. After careful consideration, we feel that it has merit but does not fully meet PLOS ONE’s publication criteria as it currently stands. Therefore, we invite you to submit a revised version of the manuscript that addresses the points raised during the review process.

Please review concerns raised by the reviewers and provide point by point response in your revised manuscript.

We look forward to receiving your revised manuscript.

Kind regards,

Muhammad Adrish

Academic Editor

PLOS ONE

Journal Requirements:

2. Thank you for including your ethics statement: 'The participants or legal guardians of minors were informed about the study and ethical acceptance was obtained from the national ethics committee of the Ministry of Health of Angola (nr.25/2020).'

a. Please amend your current ethics statement to confirm that your named institutional review board or ethics committee specifically approved this study.

3. Please provide additional details regarding participant consent. In the ethics statement in the Methods and online submission information, please ensure that you have specified (i) whether consent was informed and (ii) what type you obtained (for instance, written or verbal, and if verbal, how it was documented and witnessed). If your study included minors, state whether you obtained consent from parents or guardians. If the need for consent was waived by the ethics committee, please include this information.

4. Please include additional information regarding the survey or questionnaire used in the study and ensure that you have provided sufficient details that others could replicate the analyses. For instance, if you developed a questionnaire as part of this study and it is not under a copyright more restrictive than CC-BY, please include a copy, in both the original language and English, as Supporting Information.  If the original language is written in non-Latin characters, for example Amharic, Chinese, or Korean, please use a file format that ensures these characters are visible.

5. Please state whether you validated the questionnaire prior to testing on study participants. Please provide details regarding the validation group within the methods section.

6. We note that the grant information you provided in the ‘Funding Information’ and ‘Financial Disclosure’ sections do not match.

Reviewers' comments:

Reviewer's Responses to Questions

**Comments to the Author**

1. Is the manuscript technically sound, and do the data support the conclusions?

Reviewer #1: No

Reviewer #2: Yes

Reviewer #3: Yes

2. Has the statistical analysis been performed appropriately and rigorously? 

Reviewer #1: I Don't Know

Reviewer #2: Yes

Reviewer #3: Yes

3. Have the authors made all data underlying the findings in their manuscript fully available?

Reviewer #1: Yes

Reviewer #2: Yes

Reviewer #3: Yes

4. Is the manuscript presented in an intelligible fashion and written in standard English?

Reviewer #1: Yes

Reviewer #2: Yes

Reviewer #3: Yes

5. Review Comments to the Author

Reviewer #1: Overall this appears to be valuable data which should be shared, particularly given the paucity of data on covid from under resourced areas, but the manuscript appears to need major revisions.

Major comments:

1. The included population needs to be better defined, we need to know why people were tested. Table 2 describes this to some degree but it is not discussed in the text.

2. The conclusions of the manuscript seem to be based on the assumption that covid testing was done as a random sample survey of prevalence. What the data actually show is the positivity rate of people tested because of some, presumably, covid suspicious syndrome. There were only 5 individuals who tested positive who were listed as asymptomatic. Why were these 5 individuals tested then? We cannot conclude from these data that healthcare workers, older individuals and non-urban dwellers are at higher risk for covid. We can only conclude that when these individuals presented with an indication for covid testing such as fever/cough/myalgias, that they were more likely to test positive for covid rather than have some other cause for their symptoms. For example, it seems that younger individuals were more likely to be tested for symptoms which were not caused by covid than older individuals. Perhaps younger individuals had easier access to testing so more presented with mild, nonspecific symptoms. There are many potential explanations and confounders that make it impossible to conclude that older individuals were at higher risk of acquiring covid, though it is well established that there is higher risk of severe symptoms for older individuals.

Minor comments:

30: "screened" it does not appear that this was asymptomatic screening so I would use "tested".

41: Did you mean to say "new member of the coronavirus family"

131: With a p=3.9, there is very little value to saying that SARS-CoV-2 infection is higher in women

171: "The covid pandemic in Luanda was asymptomatic in the first three months" Do you have a reference for this?

Table 1: It may be more appropriate, because this was not a random screening but rather symptomatic testing to describe column one as "test positivity" rather than prevalence. Prevalence suggests a random survey screening.

Reviewer #2: As a preliminary paper it is fine. But this paper need some addition of some information and rearrangement.

1. Please add detail information of INIS in the introduction section and add the contribution of this institution for COVID-19. Why you chose this institute.

2. Who were those 16028 patients (line 56)? Add detail.

3. Were you take written consent from the participants for this study? add detail.

4. 1 of the positive participant had no symptom (line 110). Describe, why this asymptomatic participant was tested for COVID-19.

5. Were all participants are admitted in the hospital? what were the out come of the participants. If possible add the disease severity of all (89) participants, like severe or moderate or mild. In result section.

6. Concise the Discussion part. remove the result part from the discussion section and rearrange it.

Reviewer #3: Reviewer’s response:

The study to identify the parameters associated with SARS-CoV-2 infection in a part of Angola looks interesting to me. The findings of the paper are important for the present COVID situation. I believe that the findings would be helpful for future research in the same research arena. However, in terms of scientific context, this manuscript is well organized in some extent, although there are some drawbacks which must be addressed before being accepted to publish, such as:

A) The authors have used p values in abstract. It would be better for not using the p values in the contents of abstract.

B) The introduction part is too short. I think, the authors should include some relevant description of the sociodemographic status of Angola to provide an overview to the readers.

C) The primer and probe sequences used in the RT-PCR assay should also be mentioned in the manuscript/supplementary materials.

D) According to the lines 56-57, this ‘study was performed with 622 individuals out of 16028 individuals tested for the SARS CoV-2 infection between January to September 2020 in INIS’. On the other hand, in lines 119-120, ‘this study had the participation of 622 individuals out of a total of 16058 individuals screened for SARS-CoV-2 using RT-PCR assay at INIS’. Why this anomaly of 16028 vs 16058? And most importantly, in which basis these 622 individuals were selected for the study and others were excluded? The inclusion of all the screened population might provide more significant information regarding the disease. The authors have to justify the sample size in this regard.

E) Have the authors checked the normality of data? If so, how? Why have the authors used chi square and logistic regression for analysis? Why not other statistical tools? A justification in the methods would be helpful. Regression analysis outputs (in details) could be included as an appendix in the supplementary materials. Structured questionnaire mentioned in line 63 by the authors could be added as supplementary materials.

F) In the beginning of the description part, what is the meaning of ‘extended’ descriptive study?

G) There are some grammatical errors/misuse (for example, in the line 41, there is an additional word ‘virus’, in the line 131 ‘despite’ is not a proper word to use here, etc) prevailed throughout the manuscript which must be corrected to clearly illustrate the facts and findings of the study to the readers. Moreover, rephrasing and improvement of writing have been suggested throughout the discussion to improve the quality of the article.

6. PLOS authors have the option to publish the peer review history of their article (what does this mean?). If published, this will include your full peer review and any attached files.

Reviewer #1: **Yes: **Brett Williams

Reviewer #2: No

Reviewer #3: **Yes: **Emtiaz Ahmed

---

## [Author Response · Author response to Decision Letter 0]

15 Feb 2021

RESPONSE TO REVIEWERS AND EDITOR

“PONE-D-20-36563: Sociodemographic characteristics and risk factors related to SARS-CoV-2 infection in Luanda, Angola.”

Editor Comments:

Thank you for submitting your manuscript to PLOS ONE. After careful consideration, we feel that it has merit but does not fully meet PLOS ONE’s publication criteria as it currently stands. Therefore, we invite you to submit a revised version of the manuscript that addresses the points raised during the review process.

Please review concerns raised by the reviewers and provide point by point response in your revised manuscript.

Journal Requirements:

Author replay: The authors appreciate the comments. The manuscript was formatted according to PLOS ONE's style requirements.

2. Thank you for including your ethics statement: 'The participants or legal guardians of minors were informed about the study and ethical acceptance was obtained from the national ethics committee of the Ministry of Health of Angola (nr.25/2020).'

a. Please amend your current ethics statement to confirm that your named institutional review board or ethics committee specifically approved this study.

Author replay: The authors appreciate the comments. The ethics statement was reformulated in this version of the manuscript. The study protocol was reviewed by the scientific coordination of INIS and ethical approval was obtained from the national ethics committee of the Ministry of Health of Angola (nr.25/2020). The participants or legal guardians of minors were informed about the study and verbal consent was secured before being enrolled in the study. The information from the studied population was fully anonymized, used for the stated objectives, and kept confidential in the INIS. (lines 80 – 84)

b. Once you have amended this/these statement(s) in the Methods section of the manuscript, please add the same text to the “Ethics Statement” field of the submission form (via “Edit Submission”). For additional information about PLOS ONE ethical requirements for human subjects research, please refer to http://journals.plos.org/plosone/s/submission-guidelines#loc-human-subjects-research

Author replay: The amended statement in the Methods section was added to the Ethics Statement field of the submission form.

3. Please provide additional details regarding participant consent. In the ethics statement in the Methods and online submission information, please ensure that you have specified (i) whether consent was informed and (ii) what type you obtained (for instance, written or verbal, and if verbal, how it was documented and witnessed). If your study included minors, state whether you obtained consent from parents or guardians. If the need for consent was waived by the ethics committee, please include this information.

Author replay: The authors appreciate the comments. The additional details regarding participant consent were provided in this version of the manuscript. The study protocol was reviewed by the scientific coordination of INIS and ethical approval was obtained from the national ethics committee of the Ministry of Health of Angola (nr.25/2020). The participants or legal guardians of minors were informed about the study and verbal consent was secured before being enrolled in the study. The information from the studied population was fully anonymized, used for the stated objectives, and kept confidential in the INIS. (lines 80 – 84)

 Author replay: The authors appreciate the comments. The text was reformulated. The participants or legal guardians of minors were informed about the study and verbal consent was secured before being enrolled in the study. The information from the studied population was fully anonymized, used for the stated objectives, and kept confidential in the INIS. (lines 82 – 84)

4. Please include additional information regarding the survey or questionnaire used in the study and ensure that you have provided sufficient details that others could replicate the analyses. For instance, if you developed a questionnaire as part of this study and it is not under a copyright more restrictive than CC-BY, please include a copy, in both the original language and English, as Supporting Information. If the original language is written in non-Latin characters, for example Amharic, Chinese, or Korean, please use a file format that ensures these characters are visible.

Author replay: The authors appreciate the comments. We did not develop a questionnaire as part of this study. The questionary used in the study was prepared and made available by the national public health directorate of Angola. This information was added in this version of the manuscript. (lines 76 – 77)

5. Please state whether you validated the questionnaire prior to testing on study participants. Please provide details regarding the validation group within the methods section.

Author replay: The authors appreciate the comments. The questionary used in the study was prepared and made available by the national public health directorate of Angola. However, the questionnaire was not validated prior to testing on study participants, since the questionnaire is validated by the national public health directorate of Angola for the surveillance and investigation of SARS-CoV-2 cases in Angola. This information was added in this version of the manuscript. (lines 76 – 80)

6. We note that the grant information you provided in the ‘Funding Information’ and ‘Financial Disclosure’ sections do not match.

Author replay: The authors appreciate the comments. No specific funding was obtained for this study. This information was added in this version of the manuscript. (line 269)

Author replay: The authors appreciate the comments. No specific funding was obtained for this study and there are no correct grand numbers for the awards. This information was added in this version of the manuscript. (lines 269 – 270) The INIS is an Angolan institute of scientific research directly subordinated and supported by the Angolan Ministry of Health, whose main objective is to generate, develop and disseminate scientific, technological, and strategic knowledge about health and its determinants, for strengthening public health policies and improving the national health system in Angola (http://www.inis.ao/index.php/institucional/o-instituto). This information was added in this version of the manuscript. (lines 69 – 73)

Reviewers' comments:

5. Review Comments to the Author

Reviewer #1: Overall this appears to be valuable data which should be shared, particularly given the paucity of data on covid from under resourced areas, but the manuscript appears to need major revisions.

Author replay: The authors appreciate the comments.

Major comments:

1. The included population needs to be better defined, we need to know why people were tested. Table 2 describes this to some degree but it is not discussed in the text.

Author replay: The authors appreciate the comments. The text was reformulated. People were tested for the following reasons as follows: if they had presumably COVID-19 suspicious syndrome; if they have been in contact with someone infected; or if they traveled to any country or region with active transmission of SARS-CoV-2. This information was added in this version of the manuscript. (lines 88 – 90)

2. The conclusions of the manuscript seem to be based on the assumption that covid testing was done as a random sample survey of prevalence. What the data actually show is the positivity rate of people tested because of some, presumably, covid suspicious syndrome. There were only 5 individuals who tested positive who were listed as asymptomatic. Why were these 5 individuals tested then? We cannot conclude from these data that healthcare workers, older individuals and non-urban dwellers are at higher risk for covid. We can only conclude that when these individuals presented with an indication for covid testing such as fever/cough/myalgias, that they were more likely to test positive for covid rather than have some other cause for their symptoms. For example, it seems that younger individuals were more likely to be tested for symptoms which were not caused by covid than older individuals. Perhaps younger individuals had easier access to testing so more presented with mild, nonspecific symptoms. There are many potential explanations and confounders that make it impossible to conclude that older individuals were at higher risk of acquiring covid, though it is well established that there is higher risk of severe symptoms for older individuals.

Author replay: The authors appreciate the comments. The text was reformulated.

Minor comments:

30: "screened" it does not appear that this was asymptomatic screening so I would use "tested".

Author replay: The authors appreciate the comments. The text was reformulated. (line 30)

41: Did you mean to say "new member of the coronavirus family"

Author replay: The authors appreciate the comments. The text was reformulated. The severe acute respiratory syndrome of coronavirus 2 (SARS-CoV-2) is a new member of the family Coronaviridae. This information was added in this version of the manuscript. (lines 42 – 43)

131: With a p=3.9, there is very little value to saying that SARS-CoV-2 infection is higher in women

Author replay: The authors appreciate the comments. The text was reformulated.

171: "The covid pandemic in Luanda was asymptomatic in the first three months" Do you have a reference for this?

Author replay: The authors appreciate the comments. According to the COVID-19 daily update bulletin from the Ministry of Health of Angola, the COVID-19 pandemic in Luanda was asymptomatic in the first three months with positive patients reporting no symptoms. (lines 206 – 208). These data were obtained from the daily update of COVID-19 released by the National Directorate of Public Health of Angola. (Ministério da Saúde de Angola. Pandemia da COVID-19 em Angola. Boletin informativo 269: 16 de outubro de 2020. 2020;:58–9.)

Table 1: It may be more appropriate, because this was not a random screening but rather symptomatic testing to describe column one as "test positivity" rather than prevalence. Prevalence suggests a random survey screening.

Author replay: The authors appreciate the comments. The text in Table 1 was reformulated and the word prevalence was removed.

Reviewer #2: As a preliminary paper it is fine. But this paper need some addition of some information and rearrangement.

Author replay: The authors appreciate the comments.

1. Please add detail information of INIS in the introduction section and add the contribution of this institution for COVID-19. Why you chose this institute.

Author replay: The authors appreciate the comments. The introduction section was reformulated and information about the INIS was included in this version of the manuscript. The first cases of SARS-CoV-2 infection in Angola were detected in March 2020 at the Instituto Nacional de Investigação em Saúde (INIS), the national reference biomedical research institute located in Luanda, the capital city of Angola. (lines 49 – 51) At the beginning of the COVID-19 pandemic in Angola, the INIS was the only institution responsible for laboratory testing and surveillance of SARS-CoV-2 cases in the country. (lines 52 – 53) We chose this institution because the INIS is an Angolan institute of scientific research directly subordinated and supported by the Angolan Ministry of Health, whose main objective is to generate, develop and disseminate scientific, technological, and strategic knowledge about health and its determinants, for strengthening public health policies and improving the national health system in Angola (http://www.inis.ao/index.php/institucional/o-instituto). (lines 69 – 73)

2. Who were those 16028 patients (line 56)? Add detail.

Author replay: The authors appreciate the comments. The text was reformulated. The 16028 patients were the total number of individuals tested for SARS-CoV-2 from January to September 2020 at INIS. Angola has a weak surveillance system in which lack of epidemiological data is a major problem that being the reason why only 622 individuals with complete epidemiological data were included in this paper. This information was added in this version of the manuscript. (lines 119 – 121)

3. Were you take written consent from the participants for this study? add detail.

Author replay: The authors appreciate the comments. The participants or legal guardians of minors were informed about the study and verbal consent was secured before being enrolled in the study. The information from the studied population was fully anonymized, used for the stated objectives, and kept confidential in the INIS. This information was added in this version of the manuscript. (lines 82 – 84)

4. 1 of the positive participant had no symptom (line 110). Describe, why this asymptomatic participant was tested for COVID-19.

Author replay: The authors appreciate the comments. People were tested for the following reasons as follows: if they had presumably COVID-19 suspicious syndrome; if they have been in contact with someone infected; or if they traveled to any country or region with active transmission of SARS-CoV-2. This information was added in this version of the manuscript. (lines 88 – 90)

5. Were all participants are admitted in the hospital? what were the out come of the participants. If possible add the disease severity of all (89) participants, like severe or moderate or mild. In result section.

Author replay: The authors appreciate the comments. The text was reformulated. We agree that this data would be very important to be reported. All individuals who tested positive by RT-PCR were placed in quarantine centers established by the Angolan Ministry of Health, for clinical follow-up and isolation. Upon entering quarantine in most cases we lost track of patients and it was not possible to do a follow-up of patients, to obtain the result of the disease severity, and the clinical outcome. This information was added in this version of the manuscript. (lines 127 – 130)

6. Concise the Discussion part. remove the result part from the discussion section and rearrange it.

Author replay: The authors appreciate the comments. The text in the Discussion section was reformulated, and the result part was removed from the 

Discussion.

Reviewer #3: Reviewer’s response:

The study to identify the parameters associated with SARS-CoV-2 infection in a part of Angola looks interesting to me. The findings of the paper are important for the present COVID situation. I believe that the findings would be helpful for future research in the same research arena. However, in terms of scientific context, this manuscript is well organized in some extent, although there are some drawbacks which must be addressed before being accepted to publish, such as:

Author replay: The authors appreciate the comments.

A) The authors have used p values in abstract. It would be better for not using the p values in the contents of abstract.

Author replay: The authors appreciate the comments. The text in the abstract was reformulated and p values were removed.

B) The introduction part is too short. I think, the authors should include some relevant description of the sociodemographic status of Angola to provide an overview to the readers.

Author replay: The authors appreciate the comments. In this version of the manuscript, the text in the introduction was reformulated and the sociodemographic status of the Angolan population was included to provide an overview to the readers. Angola is a country in sub-Saharan Africa with more than 25 million inhabitants living in 18 provinces, with about 48% male inhabitants and 52% female, with an average age of 20.6 years. Luanda province is the most inhabited with about 7 million inhabitants.[9] A large proportion of the population from Luanda province lives in slums with poor basic sanitation and limited access to health care.[10] In addition, because of the oil trade, international business travelers have intensely visited the Luanda province, which could increase the chance of importing SARS-CoV-2 to the country's capital and easily spread to other regions. (lines 54 – 59)

C) The primer and probe sequences used in the RT-PCR assay should also be mentioned in the manuscript/supplementary materials.

Author replay: The authors appreciate the comments. The SARS-CoV-2 infection was screened and confirmed with real-time reverse transcriptase-polymerase chain reaction (RT-PCR) assay using the Applied Biosystems 7500 Fast RT-PCR System (Thermo Fisher Scientific), in the molecular biology laboratory of INIS, using a protocol previously described for the detection of 2019 novel Coronavirus (2019-nCoV) RNA (PCR-Fluorescence Probing) (Da an Gene, China). (lines 94 – 98) Reference: Da An Gene. Instructions for use for Detection Kit for 2019 Novel Coronavirus (2019-nCoV) RNA (PCR-Fluorescence Probing). Sun Yat-sen Univ. 2019;4:75–84. https://www.who.int/diagnostics_laboratory/eual/eul_0493_141_00_detection_kit_for_2019_ncov_rna_pcr_flourescence_probing.pdf.

D) According to the lines 56-57, this ‘study was performed with 622 individuals out of 16028 individuals tested for the SARS CoV-2 infection between January to September 2020 in INIS’. On the other hand, in lines 119-120, ‘this study had the participation of 622 individuals out of a total of 16058 individuals screened for SARS-CoV-2 using RT-PCR assay at INIS’. Why this anomaly of 16028 vs 16058? And most importantly, in which basis these 622 individuals were selected for the study and others were excluded? The inclusion of all the screened population might provide more significant information regarding the disease. The authors have to justify the sample size in this regard.

Author replay: The authors appreciate the comments. The text was reformulated in this version of the manuscript. The 16028 patients were the total number of individuals tested for SARS-CoV-2 from January to September 2020 at INIS. Angola has a weak surveillance system in which lack of epidemiological data is a major problem that being the reason why only 622 individuals with complete epidemiological data were included in this paper. (line 119 – 121) People were tested for the following reasons as follows: if they had presumably COVID-19 suspicious syndrome; if they have been in contact with someone infected; or if they traveled to any country or region with active transmission of SARS-CoV-2. (lines 88 – 90)

E) Have the authors checked the normality of data? If so, how? Why have the authors used chi square and logistic regression for analysis? Why not other statistical tools? A justification in the methods would be helpful. Regression analysis outputs (in details) could be included as an appendix in the supplementary materials. Structured questionnaire mentioned in line 63 by the authors could be added as supplementary materials.

Author replay: The authors appreciate the comments. The text in statistical analysis was reformulated. The normality of data distribution was checked using the values of skewness and kurtosis. Categorical variables were presented as frequencies and percentages, while continuous variables with the data normally distributed were presented as mean and standard deviation (SD). Chi-square (X2) test was used to compare frequencies and identify the relationship between categorical variables. Besides that, logistic regression analysis and odds ratio (OR) with their 95% confidence intervals (CIs) were calculated to determine the strength and direction of the interaction between variables. This information was added in this version of the manuscript. (lines 108 – 114)

F) In the beginning of the description part, what is the meaning of ‘extended’ descriptive study?

Author replay: The authors appreciate the comments. The text was reformulated and the word “extended” was removed. 

G) There are some grammatical errors/misuse (for example, in the line 41, there is an additional word ‘virus’, in the line 131 ‘despite’ is not a proper word to use here, etc) prevailed throughout the manuscript which must be corrected to clearly illustrate the facts and findings of the study to the readers. Moreover, rephrasing and improvement of writing have been suggested throughout the discussion to improve the quality of the article.

Author replay: The authors appreciate the comments. The text was reformulated and some grammatical errors were checked.

---

## [Decision Letter · Decision Letter 1]

15 Mar 2021

Sociodemographic characteristics and risk factors related to SARS-CoV-2 infection in Luanda, Angola

PONE-D-20-36563R1

Dear Dr. Cruz dos Santos Sebastião,

We’re pleased to inform you that your manuscript has been judged scientifically suitable for publication and will be formally accepted for publication once it meets all outstanding technical requirements.

Kind regards,

Muhammad Adrish, MD, MBA, FCCP, FCCM

Academic Editor

PLOS ONE

Additional Editor Comments (optional):

All queries have been answered by the authors

Reviewers' comments:

Reviewer's Responses to Questions

**Comments to the Author**

1. If the authors have adequately addressed your comments raised in a previous round of review and you feel that this manuscript is now acceptable for publication, you may indicate that here to bypass the “Comments to the Author” section, enter your conflict of interest statement in the “Confidential to Editor” section, and submit your "Accept" recommendation.

Reviewer #2: All comments have been addressed

Reviewer #3: All comments have been addressed

2. Is the manuscript technically sound, and do the data support the conclusions?

Reviewer #2: Yes

Reviewer #3: Yes

3. Has the statistical analysis been performed appropriately and rigorously? 

Reviewer #2: Yes

Reviewer #3: Yes

4. Have the authors made all data underlying the findings in their manuscript fully available?

Reviewer #2: Yes

Reviewer #3: Yes

5. Is the manuscript presented in an intelligible fashion and written in standard English?

Reviewer #2: Yes

Reviewer #3: Yes

6. Review Comments to the Author

Reviewer #2: This paper is well written and preliminary information of the country was described. Will waiting for the next upcoming information in next paper.

Reviewer #3: Thanks to the authors for properly addressing the questions raised. However, I strongly recommend the authors to recheck carefully the reference styles as there are a few anomalies (for instance, Ref. No. 13 has additional pdf. link which is inappropriate). Please follow the PLOS ONE referencing style guidelines.

7. PLOS authors have the option to publish the peer review history of their article (what does this mean?). If published, this will include your full peer review and any attached files.

Reviewer #2: No

Reviewer #3: **Yes: **Emtiaz Ahmed

---

## [Editor Report · Acceptance letter]

17 Mar 2021

PONE-D-20-36563R1 

Sociodemographic characteristics and risk factors related to SARS-CoV-2 infection in Luanda, Angola 

Dear Dr. Sebastião:

I'm pleased to inform you that your manuscript has been deemed suitable for publication in PLOS ONE. Congratulations! Your manuscript is now with our production department. 

Kind regards, 

on behalf of

Dr. Muhammad Adrish 

Academic Editor

PLOS ONE